Livestock as a potential biological control agent for an invasive wetland plant

Silliman Brian R. 1 Brian.Silliman@duke.edu
Mozdzer Thomas 2
Angelini Christine 3
Brundage Jennifer E. 4
Esselink Peter 5 6
Bakker Jan P. 5
Gedan Keryn B. 7
van de Koppel Johan 5 8
Baldwin Andrew H. 4
1 Division of Marine Science and Conservation, Nicholas School of the Environment, Duke University , Beaufort, NC , USA
2 Department of Biology, Bryn Mawr College , Bryn Mawr, PA , USA
3 Department of Environmental Engineering Sciences, University of Florida , Gainesville, FL , USA
4 Department of Environmental Science and Technology, University of Maryland , College Park, MD , USA
5 Community and Conservation Ecology, University of Groningen , Groningen , The Netherlands
6 PUCCIMAR Ecological Research and Consultancy , The Netherlands
7 Department of Biology, University of Maryland , College Park, MD , USA
8 Spatial Ecology Department, Royal Netherlands Institute for Sea Research (NIOZ) , Yerseke , The Netherlands
Yoccoz Nigel
Electronic publication date: 2014 Sep 23
Publication date: 2014
Volume: 2
Electronic Location ID: e567
Received 2014 Apr 20; Accepted 2014 Aug 18
Copyright: © 2014 Silliman et al.
Copyright year: 2014
Copyright holder: Silliman et al.
License: This is an open access article distributed under the terms of the Creative Commons Attribution License, which permits unrestricted use, distribution, reproduction and adaptation in any medium and for any purpose provided that it is properly attributed. For attribution, the original author(s), title, publication source (PeerJ) and either DOI or URL of the article must be cited.
License URL: https://creativecommons.org/licenses/by/4.0/

Keywords: Top-down control, Salt marshes, Invasive species, Biocontrol

Funding: Netherlands Royal Academy of Arts and Sciences Maryland Agricultural Experiment Station Support for this study was provided by a fellowship to Brian Silliman from the Netherlands Royal Academy of Arts and Sciences and a grant to Andrew Baldwin from the Maryland Agricultural Experiment Station. The funders had no role in study design, data collection and analysis, decision to publish, or preparation of the manuscript.

==============================
Invasive species threaten biodiversity and incur costs exceeding billions of US$. Eradication efforts, however, are nearly always unsuccessful. Throughout much of North America, land managers have used expensive, and ultimately ineffective, techniques to combat invasive Phragmites australis in marshes. Here, we reveal that Phragmites may potentially be controlled by employing an affordable measure from its native European range: livestock grazing. Experimental field tests demonstrate that rotational goat grazing (where goats have no choice but to graze Phragmites) can reduce Phragmites cover from 100 to 20% and that cows and horses also readily consume this plant. These results, combined with the fact that Europeans have suppressed Phragmites through seasonal livestock grazing for 6,000 years, suggest Phragmites management can shift to include more economical and effective top-down control strategies. More generally, these findings support an emerging paradigm shift in conservation from high-cost eradication to economically sustainable control of dominant invasive species.

Introduction

Invasive species globally threaten biodiversity and, in the United States alone, incur costs to human economies estimated to exceed 120 billion US$ each year (Pimentel, Zuniga & Morrison, 2005). Controlling the spread and reducing the impacts of invasive species are therefore foundational objectives of conservation science and policy (Kareiva & Marvier, 2010). Historically, eradication of invasive species has been an ideal goal of management programs, but this has rarely been achieved on ecologically relevant spatial or temporal scales (Kettenring & Adams, 2011). In the majority of cases, complete and permanent removal of these species is simply unrealistic (Sax, Stachowicz & Gaines, 2005). Consequently, the objectives of invasive species’ management are being recast to prioritize control and mitigation, rather than elimination, of invasive species’ impacts. In addition, conservation groups are becoming increasingly focused on finding solutions that not only achieve their goals but also bolster local economies (Kareiva & Marvier, 2010). Win–win synergisms of this type, however, are rare. To ensure long-term efficacy of control-oriented programs, management strategies should be tailored to both local habitat requirements (i.e., duration, frequency and intensity of control measures) and community needs.

Invasive plants that form expansive monocultures are often key targets for management due to the direct, and usually negative, impact they have on ecosystem structure, function, and services (e.g., cordgrass: Neira et al., 2006, crested wheatgrass: Christian & Wilson, 1999, reed canary grass: Lavergne & Molofsky, 2004, Japanese stiltgrass: Flory & Clay, 2010). In the United States, control of invasive plants has been attempted through herbicide application, mechanical removal (e.g., mowing, burning, excavation), or biological control programs that are often costly to implement, difficult to sustain over sufficient timescales, or may result in unintended, harmful consequences (e.g., spillover of herbicides, non-target impacts of arthropod control agents, landscape damage; see Kettenring & Adams, 2011 for review). In Europe, however, farmers have been culling (whether intentionally or not) similarly ‘invasive’ plants long before such modern control techniques by deploying livestock to feed on dense vegetation. Grazing by large-bodied domestic herbivores, such as cows, horses, sheep, and goats, cannot only be effective in suppressing dominant plants (Esselink et al., 2000), but can also result in reciprocal positive effects for humans by generating valuable goods, including meat, milk, leather, and wool to support local economies. In the United States, use of livestock to control invasive species has been largely restricted to terrestrial grasslands where this method has met with mixed success (e.g., DiTomaso, 2000; Reiner & Craig, 2011, but see Marty, 2005; Tesauro & Ehrenfeld, 2007 for wetlands). Low species richness and discrete plant zonation common in wetland ecosystems may allow for greater success and targeted control of invasive plants by livestock. New evidence from North American and European marshes, which we document below, coupled with prior research on long-term grazing impacts on plant distribution in Europe, suggests that livestock can be a cost-effective tool for managing the impacts and spread of monoculture-forming invasive plants in wetlands, where monoculture-forming invasive plant species are common and drive large-scale ecosystem change (Zedler & Kercher, 2004).

Under natural field settings, there is broad support for the ability of herbivores to suppress invasive plant success. Specifically, herbivores can reduce invasion success by limiting both invasive plant establishment and performance (Levine, Adler & Yelenik, 2004), with generalist native herbivores strongly suppressing invasive plants (Parker, Burkepile & Hayt, 2006; Morrison & Hay, 2011). Invasive herbivores, on the other hand, have opposite effects and can facilitate invasions by reducing the abundance of native species (Parker, Burkepile & Hayt, 2006). This suggests that co-evolution/exposure of both herbivore and autotroph are important considerations when choosing an appropriate biocontrol agent. A novel management approach to maximize potential for success would attempt to control an invasive plant with an introduced herbivore (e.g., domestic livestock) that has a demonstrated effect in controlling the plants in their native range.

In Eastern North American marshes, the common reed, Phragmites australis (Cav.) Trin. ex Steud. has invaded with unrelenting success since its cryptic introduction during the 18th century from Europe (Chambers, Meyerson & Saltonstall, 1999; Saltonstall, 2002). Phragmites, which reaches average heights of >3 m and generates dense layers of lignified litter, outcompetes native plants for light and often forms expansive, towering monocultures (Bertness, Ewanchuk & Silliman, 2002; Silliman & Bertness, 2004; Meyerson, Saltonstall & Chambers, 2009). Introduced Phragmites is particularly successful in marshes along developed shorelines (Bertness, Ewanchuk & Silliman, 2002; Silliman & Bertness, 2004; King et al., 2007), and its rate of invasion is likely to increase in the future with predicted increases in anthropogenic N pollution and rising CO2 concentrations (Bertness, Ewanchuk & Silliman, 2002; Mozdzer & Zieman, 2010; Mozdzer & Megonigal, 2012; Mozdzer, Brisson & Hazelton, 2013). Where Phragmites has become established, native plant diversity declines precipitously (Silliman & Bertness, 2004), ecosystem processes such as nitrogen cycling, methane emissions, and accretion change (e.g., Rooth, Stevenson & Cornwell, 2003; Windham & Ehrenfeld, 2003; Mozdzer & Megonigal, 2013), and once-expansive marsh vistas become obfuscated by this impenetrable grass.

The control of Phragmites has dominated marsh conservation efforts in the Northeastern US for the past 30 years (Hazelton et al., 2014). During this time, no cost-effective, long-term control measures have been found. For example, land managers and private organizations have treated over 80,000 hectares of marsh with herbicide over the past five years with limited success, despite costs that exceed $4.6 million per year (Martin & Blossey, 2013). Similarly, mechanical removal techniques, such as mowing and burning, have proven to be uneconomical, given their high labor costs, and ineffective (Lee, 1990; Cowie et al., 1992). While insect biocontrol has been investigated (Tewksbury et al., 2002; Van Driesche et al., 2002), and specific biocontrol agents tested in laboratory conditions (e.g., Lambert, Winiarski & Casagrande, 2007), it is currently not an option available to land managers, in part because some prospective control agents do greater damage to native strains of Phragmites than the invasive (e.g., Lambert, Winiarski & Casagrande, 2007).

As a facultative halophyte, Phragmites distribution is largely restricted by salinity in Europe and North America. Phragmites salinity tolerance may exceed normal seawater (∼33ppt) (Chambers et al., 2003), but its competitiveness increases with decreasing salinity in the high marsh elevations associated with freshwater seepage or in naturally brackish wetlands (Minchinton & Bertness, 2003). However, within brackish marshes in Europe, Phragmites abundance is markedly limited. For instance, in the 400-km2 salt marshes of the Wadden Sea, Phragmites accounts for only 2.5% of vegetation coverage (Esselink et al., 2009), a significant decrease from its historical extent. Although nutrient pollution has been attributed to Phragmites die back in Europe (Van Der Putten, 1997), two primary reasons likely account for Phragmites’ reduced distribution in European brackish marshes: (1) seawall construction and land reclamation during the Middle Ages (c. A.D. 1000–1300), which caused major loss of brackish marshes with Phragmites and (2) an extended history of livestock grazing in these marshes. For example, along the brackish coast of the microtidal Baltic, reed beds dominated by Phragmites were transformed into salt meadows from c. 4000 B.C. onwards, when human exploitation started (Vestergaard, 1998). Likewise, in the Netherlands, marshes have been used as rangelands since Late Neolithic, i.e., 3500 B.C. In these heavily grazed European marshes, Phragmites is rare, but in areas where livestock grazing has been abandoned in recent times, Phragmites has become dominant again (Dijkema, 1990; Jutila, 1999; Esselink, Fresco & Dijkema, 2002; Esselink et al., 2009). A recent study confirmed these observational results: specifically, in marshes still grazed by livestock (Sammul, Kauer & Koster, 2012), Phragmites has increased in relative abundance inside grazer exclusion cages but not in control, grazed areas (Milotic et al., 2010; Esselink, 2008, unpublished data). It is unclear whether similar top-down control methods would be effective in mitigating the impacts of introduced Phragmites in North America, and, if so, which large grazers and deployment strategy would result in an ecologically effective and economically sustainable solution for both land managers and farmers.

Based on observational and experimental evidence revealing that top-down forces limit Phragmites in its native range in Europe, we explored the potential for livestock control of invasive Phragmites in North America, and, reciprocally, the nutritional benefit of Phragmites to livestock consuming it. Our specific objectives were: (1) to evaluate the ability of caged livestock to control invasive Phragmites and increase plant biodiversity in a small-scale experimental setting, (2) to test if various commercially reared livestock breeds will readily consume Phragmites, and (3) to investigate the nutritional value of Phragmites as livestock forage.

We tested the potential for livestock to control introduced Phragmites with a goat inclusion field experiment in a Phragmites-invaded North American marsh. As top-down control of invasive plants by free-ranging livestock can be hindered by alternative grazing options that could be superior in quality (Belovsky, 1986; Vulink & Drost, 1991a; Vulink, 2001), we chose to conduct a pressed, grazing experiment in a marsh already dominated by Phragmites. To evaluate the potential of additional top-down control agents besides goats and the nutritional benefits of Phragmites to livestock, we conducted no-choice feeding trials with cows and horses and, using data from past, unpublished studies, assessed whether livestock can digest Phragmites effectively and if Phragmites nutritional quality varies significantly over a growing season.

Methods

To experimentally test the hypothesis that livestock can suppress Phragmites monocultures in North American marshes and promote the recovery of native plants, we established randomly located replicated (n = 4) goat enclosures (8.5 m × 40 m) made out of wire fencing, a single-strand of electrical wire, and metal stakes in a Phragmites-dominated freshwater marsh in the Beltsville Agricultural Research Center in Beltsville, Maryland, USA. Two domestic goats (IACUC number 103453) were stocked in each of the enclosures (a stocking rate of 58.8 goats/ha), which were paired with ungrazed control plots (also 8.5 m × 40 m) for three treatment periods of 2–4 weeks, beginning mid September 2008, late May 2009, and late August 2009. Two goats per enclosure were used because, first, goats are social animals and solitary confinement might alter their well-being and grazing behavior, and second, because >2 goats would result in too rapid consumption of available plants for grazing (W Hare, pers. comm., 2012, Veterinarian, USDA Beltsville Agricultural Research Center). Goats were left in enclosures until Phragmites was completely consumed within at least one of the four enclosures to maximize the duration of grazing and preventing starvation of goats. Phragmites was allowed to re-sprout and grow to a height of about 1.5 m before applying the next round of grazing. This approach was implemented to allow potential colonization of native plants and maximize depletion of belowground resources of Phragmites (i.e., a level of grazing pressure in excess of that typical of standard rotational grazing practices was intended). Phragmites stem density and height of the five tallest stems were measured before and after each grazing period in four 1-m2 quadrats spaced systematically at 4, 8, 12, and 16 m along the center line of each grazed and control plot (total = 32 quadrats). Percent cover and plant species richness and diversity were determined in a 100-m2 (5 m × 20 m) “module” centered within each grazed and control plot (Peet, Wentworth & White, 1998; Gurevich, Scheiner & Fox, 2006). Plants that were seedlings or lacking flowering or fruiting material, but that could be distinguished as separate species, were identified as “morphospecies” and used in calculations of richness and diversity. Nativity (native or introduced) was determined using the USDA PLANTS database for plants identified to species level, or for taxa where all species were native or introduced. Comparisons between grazed and ungrazed plots were made using mixed model repeated measures ANOVA (n = 4) using the MIXED procedure of SAS 9.2 (SAS Institute, Cary, North Carolina), with each pair of grazed and ungrazed plots treated as a block, after checking assumptions of homogeneity of variances and normality. Stem density and stem height from the four 1-m2 quadrats within each plot were averaged to generate a single value for each grazed and control plot prior to analysis. Simple effect slices were used to test for significant grazing effects for each sampling date, and the Kenward–Roger method used to calculate denominator degrees of freedom (which can result in fractional ddf).

To explore whether Phragmites could be restricted by other domestic livestock species in addition to goats, we conducted no-choice feeding trials in August of 2011 with freshly cut 30-cm sections of Phragmites stems and leaves from established reed stands (>2 m in diameter). To do so, we offered a 30-cm section of Phragmites stem to 20 individual horses and cows and counted the number of individuals who fully consumed the section after 30 s.

To explore the nutritional quality of Phragmites relative to five other common marsh plants, we present data here from a past, unpublished study that asked this question and measured temporal variation in leaf quality of six marsh plants over a growing season. Specifically, both crude-protein and energy content were assessed based on sampling of the top five leaves of each species at each sampling date. Samples with a fresh weight of >200–300 g were collected every 3–4 weeks throughout the 1992 grazing season (∼end of May–mid September) from a brackish salt marsh in Dollard Bay (53°16′N, 7°10′E), the Netherlands (Marsh Section 3 of the study area in Esselink et al., 2000). Crude protein content was calculated by multiplying the nitrogen content by a factor 6.25 (Allen, 1989). The in-vitro digestible dry matter content (DDM) was used as parameter for the energy content of the plant material from the animal perspective. DDM was measured according to Tilley & Terry (1963).

Results

By the end of our livestock enclosure experiment, goats had strongly suppressed Phragmites growth, reducing stem density by ∼50% (29–14 stems m-2; Fig. 1A), stem height by ∼60% (3.9–1.4 m; Fig. 1B), and percent cover five-fold (from 94% to 21%; Fig. 1C). In grazed plots, Phragmites resprouted from rhizomes or colonized from adjacent ungrazed plots, but never attained the stem density, height, or cover of that in ungrazed plots, except in spring of the second growing season during early shoot emergence (Fig. 1). Concomitant with the strong decline of Phragmites was a marked and significant increase in plant species richness and Shannon–Weiner diversity index by the end of the experiment (∼100% & 400% respectively, Figs. 2A and 2B). At the end of the experiment, goat-grazed plots contained a total of 36 taxa (22 confirmed native species, 8 indeterminate, and 6 introduced taxa), while ungrazed plots contained only 20 taxa (12 native species. 2 indeterminate, and 6 introduced taxa), all of which were at low abundance relative to Phragmites. Important native species that colonized (from seeds in the seed bank or dispersed to the site) included Alisma subcordatum, Epilobium coloratum, Leersia oryzoides, Mimulus ringens, Penthorum sedoides, Polygonum punctatum, and Ranunculus sceleratus.

Figure 1 Goat grazing impacts on Phragmites.

Effect of goats on three measures of Phragmites australis abundance from July 2008–October 2009. Values are mean ± SE for 4 replicate enclosures (grazed) and control (ungrazed) plots. Stem density and height were determined in 1-m2 plots and percent cover was determined in 100-m2 plots. Arrows indicate the initiation of grazing periods; for cover (C), the third grazing period falls between the last two measurement points. Results of repeated measures ANOVA are given within each panel for effects of grazing (G), date (D), and their interaction (G × D). + P < 0.1, ∗P < 0.05, ∗∗P < 0.01, ∗∗∗P < 001, and ∗∗∗∗P < 0.0001. Asterisks above plotted points denote a significant grazing effect for that sampling date (P < 0.05, simple effect of grazing by date); P-value given for the last stem density comparison, where P was between 0.05 and 0.1.

Figure 2 Goat grazing impacts on plant diversity.

Changes in plant species richness (A) and Shannon–Weaver diversity (B) throughout the experiment in response to rotational goat grazing. Values are means ± SE for 4 replicate enclosures (grazed) and controls (ungrazed). Arrows indicate grazing period between sampling events; the third grazing period falls between the last two measurement points. An asterisk indicates a significant difference between grazed and ungrazed plots on a particular date (P < 0.05, simple effect of grazing by date). Also presented are results of repeated measures ANOVA for effects of grazing (G), date (D), and their interaction (G × D). ∗P < 0.05, ∗∗P < 0.01, ∗∗∗P < 001, and ∗∗∗∗P < 0.0001.

In our no-choice feeding trials, both horses and cows readily ate Phragmites (20 out of 20 individuals for both species ate the 30 cm stem section offered). In comparison with five other marsh plant species, nutritional quality of Phragmites had lower digestible matter content but higher protein content (Fig. 3A). It must be noted that North American introduced Phragmites was introduced from Europe, and we do not expect there to be any differences in tissue quality. Plants described in Fig. 3A have congeneric representation in North American wetlands, and serve as our proxy for comparable nutritional quality. Throughout the 4-month grazing season in the Dollard salt marshes, crude-protein content in Phragmites leaves was surprisingly high. The energy content of Phragmites leaves, on the contrary, was lower than in other common plant species, and dropped markedly during the course of the grazing season (Fig. 3B); after ∼mid-July it fell below the level of maintenance requirement for cattle (Australian Research Council, 1980; Van Soest, 1982). These values for digestible dry matter were within the range found by a larger survey of Dutch plant species palatability to cattle, which found Phragmites to be an important natural forage species (Bokdam & Wallis de Vries, 1992).

Discussion

Our results and those of others in Europe indicate that controlling invasive Phragmites in North America via purposeful livestock grazing has a high potential to suppress its impact on native plant communities. Our feeding trial from The Netherlands, together with evidence from livestock-removal and comparative studies in European marshes (Esselink, Fresco & Dijkema, 2002), suggest that livestock strongly restrict Phragmites distribution and facilitate the growth of shorter grasses and forbs in its native habitat. These results, in combination with our goat enclosure experiment in the U.S., indicate Phragmites is also likely susceptible to top-down control by livestock in Eastern North America. Furthermore, the short-term duration of our goat inclusion periods (3, <1 month deployments over 1 year), affordable infrastructure (wire fences), and limited number of animals (2 goats per 340-m2 plot) needed to reduce Phragmites cover, imply that livestock has the potential to offer an effective, pesticide-free solution for managers trying to regulate this invasive plant, and likely other invasive plants that form vast monocultures. The conclusion that goat grazing could be an economically sustainable, win–win invasive plant control solution is also supported by the fact that livestock can persist over short time periods (i.e., weeks to months) on Phragmites-based diets without detriment to their health.

In invaded areas, Phragmites outcompetes native plants for light and space due to its height, dense canopy, thick litter, and rapidly growing rhizomes, and these advantages drive its rapid expansion and dominance across marshes (Bertness, Ewanchuk & Silliman, 2002; Silliman & Bertness, 2004; Mozdzer & Zieman, 2010; Holdredge et al., 2010). Our results and prior studies from Europe indicate that domestic livestock can reduce the competitive advantage of Phragmites through a combination of eating down or trampling live stems, breaking up the litter mat, and severing rhizomes with their hooves (Turner, 1987). Combined, these activities can increase the light availability to native plants, reduce belowground competition for nutrients, and thus provide opportunities for recolonization of native plants, estuarine nekton, and even endangered turtles (Angradi, Hagan & Able, 2001; Tesauro, 2002; Hunter et al., 2006; Tesauro & Ehrenfeld, 2007; Tesauro, 2002). In disrupting Phragmites growth, livestock also have the potential to reduce seed set, an important mechanism of expansion of Phragmites in North America (McCormick et al., 2010). By removing the primary mechanisms of Phragmites competitive exclusion (i.e., its height and litter), livestock may not only facilitate recovery of native plants and dependent faunal communities (e.g., invertebrates, arthropod herbivores), but also restore coastal ecosystem services. However, we must caution that introduction of livestock to invaded marshes in North America will not lead to a complete return to the pre-invasion marsh structure. Instead, we suspect that an alternative state will be induced (Hobbs & Norton, 1996). Such an alternative state may be characterized by a reduced Phragmites dominance and an increased abundance of native plants and fauna. Livestock grazing, however, is not without its own effects on ecosystem characteristics, affecting soil bulk density, soil organic matter, mineralization rate (Schrama et al., 2013), invertebrate abundance, among others. Comparative, multi-year trials are needed to assess grazing impacts and to determine the best regimen of grazers for Phragmites control, ecosystem integrity, and livestock production.

Context-dependency of grazer control and next steps

Evidence from our study coupled with other livestock and large grazer manipulative experiments (Tesauro, 2002; Sturm, 2007; URS, 2005) suggest that the efficacy of livestock control of Phragmites in North America will be context-dependent and contingent both on the grazing regime and the background cover of Phragmites. Specifically, these studies suggest that livestock can control Phragmites when its cover is high and livestock are forced to graze in those areas (Fig. 4). For example, when Phragmites is dominant and grazers are enclosed in these areas as in our experiment and a 2-year study in New Jersey, USA that manipulated goats and sheep in small (0.8 ha), un-replicated pens (Tesauro, 2002), livestock were effective at reducing Phragmites from ∼100% to <50% cover. In contrast, when Phragmites is uncommon and livestock are free-roaming (i.e., grazers not forced to eat Phragmites only), horses and deer in Maryland increased Phragmites abundance relative to grazer exclusion plots (Sturm, 2007). Similarly, goats released into larger Phragmites-invaded tidal marsh in New Jersey did not reduce Phragmites cover and consumed other marsh plants to a greater degree (URS, 2005). These findings suggest that if livestock are released into mixed marsh plant communities where alternative food choices are abundant (i.e., Phragmites is uncommon) large grazers have the potential to facilitate Phragmites invasion, and thus be counterproductive to management objectives. This conclusion is supported by our nutritional content study (Fig. 3) and those of others (e.g., Vulink & Drost, 1991b) that indicate Phragmites has lower digestible matter content (although higher protein content) than other common European salt marsh plants relative and thus would not likely be preferred by grazers if given a choice.

Figure 3 Marsh plant nutritional value.

(A) Comparison of nutritional quality among six potential food plants in the cattle-grazed Dollard salt marshes, Netherlands. Figure shows the in vitro digestible dry-matter (DDM) content plotted with the crude-protein content (mean ± SD) in young leaf tissue (five top leaves) during the grazing season (3rd decade of May–mid September). (B) Fall of forage quality (i.e., energy content) in leaf tissue of primary shoots of Phragmites in the Dollard salt marshes, Netherlands, during the grazing season based on the in vitro digestible dry-matter content in leaf tissue and compared with the level that cattle require for maintenance (after Australian Research Council, 1980; Van Soest, 1982).

Figure 4 Images of goat grazing impacts.

Pictures of impacts of no-choice goat grazing in the Phragmites-dominated experimental wetland.

Based on these conclusions, we suggest preliminary guidance for applications of livestock for invasive plant control (Table 1) and recommend future directions for research. Our grazing experiment, in which grazing had stronger effects in early summer than late summer, as well as our assessment of a decline in Phragmites’ nutritional value through time, indicate that the timing of grazer implementation may be critical for the success of livestock control programs as young stems have higher nutritional quality (Fig. 3) and grazing on young Phragmites’ stems in early spring is more effective at reducing future regrowth and promoting native plant recovery (Karunaratne, Asaeda & Yutani, 2004). Future research should address whether springtime or early summer grazing has stronger impact on Phragmites and other monoculture forming invasive plants. Looking forward, the next step in determining the potential for livestock to control Phragmites and facilitate the recovery of native plants, animals and pre-invasion soil properties is to test these ideas at larger scales and over multiple years to compare to reference wetlands without grazer control of invasive plants and those using other invasive species control techniques.

Table 1 Management considerations.

Based on our experimental findings, we find that livestock grazing for control of invasive plants holds great potential to reduce invasive plant biomass, increase plant diversity, and support livestock production. For effective control and to avoid negative impacts of over-grazing, we recommend:	
1. High-intensity, short-duration, rotational grazing. Grazers will be most effective in dense, monotypic stands that are common in the establishment and spread phases of invasions. Periods without grazers are likely very important in allowing native plants to establish (Fig. 2) and for the health of grazing livestock (Fig. 3B).	
2. Small scale enclosures to concentrate feeding on the dominant, invasive plant (Fig. 4). In the case of Phragmites, the high digestible dry matter content of other wetland plant species (Fig. 3A) suggests that livestock permitted to graze freely might prefer other available plants.	
3. The incorporation of grazing into a long-term management scheme. Grazing will not eradicate an invasive plant, but will release native plants from invasive dominance temporarily. Therefore, grazing may have to occur throughout many years, and possibly indefinitely.	
4. Species-specific grazing windows. Time grazing events to suppress dominant plant invaders (i.e., early in the growing season) and limit clonal regrowth while providing adequate windows for native plants recolonization.	
5. Landscape considerations. Grazing is unlikely to be effective in soft-sediment environments, such as low elevation marshes, where trampling effects may overwhelm native plant recovery. Grazers will be most effective on firm soils, such as those in the high marsh and at the upland marsh ecotone, where Phragmites invasions begin.	

Finally, prior to application, it is critical to investigate the potential for livestock grazing impacts on non-target organisms and ecosystem processes. Decisions about the placement and timing of grazers should incorporate local site knowledge to avoid priority seasons and habitat areas for nesting birds and other possibly sensitive taxa or conservation targets. Further research could also identify the effects of short periods of grazing on critical wetland ecosystem processes such as soil compaction and surface elevation accretion, and examine the possibility that invasive plant seeds remain viable during livestock gut passage and are unwittingly dispersed to other sites.

Although inter-site variation and inter-annual differences make the synthesis of experimental findings from different decades and continents difficult, we find the consistent palatability of Phragmites to a diverse set of commercially important grazers in Europe and North America inspiring to pursue livestock grazing as a invasive species management tool. Other effective methods may be found by looking across ecosystems and continents for scenarios where dominant plants, whether purposefully or not, are being controlled using measures that involve and benefit local communities.

For monoculture-forming plants invading softer, lower elevations of marshes, such as Spartina alterniflora, on the West Coast of the US and China or Cuelerpa in soft-sediment intertidal expanses throughout the world, domestic livestock are not likely an option for management. However, other economically sustainable, but currently overlooked, rotational top-down control methods may work for these species, such as systematic human harvesting of invasive plants to be used as livestock feed or biofuel.

Evidence from European marshes that a top-down restoration strategy will work

Restoration of coastal marshes presently dominated by Phragmites has not been practiced widely in Europe, except for in the Baltics. There, a comparison of uninterruptedly managed (seasonal summer grazing), abandoned (no grazing) and restored (i.e., summer grazing re-introduced after abandonment) sites in coastal marshes revealed that plant biomass in restored sites rapidly changed back to the level of managed marshes (Sammul, Kauer & Koster, 2012) and Phragmites cover decreased significantly. Plant species composition remained different, but typical coastal grassland species colonized and increased in abundance in restored sites (Sammul, Kauer & Koster, 2012). The response of soil properties to the re-introduction of grazing evolved more slowly. In abandoned sites for instance, organic matter content and C/N ratio were significantly higher and bulk density significantly lower than in continuously managed sites. In the five-year-old restored sites, however, all soil variables still did not differ from abandoned sites, implying that the results of grazer-driven restoration may be slow for some variables. In addition, return of tall-growing Phragmites is likely if management intensity wanes. Sammul, Kauer & Koster (2012) conclude that Phragmites can indeed be suppressed in sites where it is dominant, but considering the slow response of soil properties, long-lasting periods of livestock-enhanced restoration should be planned in order to reach pre-abandonment environmental conditions.

Further incentives for integrating top-down control into invasive species management

While we have specifically focused on the control of an invasive plant as a management objective, this control has ancillary benefits and indirectly addresses multiple conservation targets. In addition to offering a solution for management of invasive plants that form expansive, hard-to-eradicate monocultures, livestock control programs can have reciprocal, positive impacts on local economies. Specifically, as is done in Europe, farmers could potentially receive payment for their services in controlling invasive species, and resources (e.g., fencing, transportation of livestock) to engage in such programs. At the same time, conservation groups and government organizations will receive more cost-effective and ecologically friendly tools to manage problematic invasive plants. Several instances of companies offering services of goats and other livestock to control Phragmites in urban wetlands in New York City, tidal wetlands in the Chesapeake Bay (e.g., Eco-Goats), and riverbank wetlands in the U.S. mid-west suggest that these ventures are marketable and can suppress Phragmites success (A Deer, pers. comm., 2014). More data are needed to confirm the short- and long-term sustainability of these business models. In many arenas, win–win solutions of economic gains in controlling invasives are often criticized with the argument that at some point the invasive species is going to be needed to maintain the economic model based upon it. In our proposed scenario using goat control of an invasive plant, however, we do not believe this will ever be the case as goats will likely always have more invasive to graze in the area (e.g., Kudzu) and, even when invasive plants have been locally suppressed, goats can still provide numerous benefits to their owners (e.g., dairy and meat production).

Beyond the target site where grazing is implemented, control of Phragmites reduces propagule pressure (McCormick et al., 2010) and interrupts positive feedback reducing spread to un-invaded sites (Hazelton et al., 2014). In addition, this approach provides an alternative treatment option when herbicide use is unacceptable or infeasible or where reduced Phragmites biomass and some native cover is an acceptable goal.

This general framework, designed to link invasive species management with the production of useable goods and benefit of local economies can also be applied to other systems where invasive species threaten ecosystem services (Tulbure, Ghioca & Whigham, 2007; Levin, 2006). By identifying, and then harnessing the positive effects of grazers, coastal managers could potentially fulfill their conservation goals with significant reduction in cost. Overall, a shift in invasive species management from eradication to mitigation of invasive species impacts is creating opportunities for the implementation of new strategies, including the use of atypical top-down control agents.

The manuscript benefited from comments from Sip van Wieren on an earlier draft of the manuscript.

Additional Information and Declarations

Competing Interests

Author Contributions

Animal Ethics

Jennifer E. Brundage is employed by the EPA; Peter Esselink is employed by PUCCIMAR Ecological Research and Consultancy.

Brian R. Silliman analyzed the data, wrote the paper, prepared figures and/or tables, reviewed drafts of the paper.

Thomas Mozdzer, Christine Angelini, Jan P. Bakker and Johan van de Koppel wrote the paper, reviewed drafts of the paper.

Jennifer E. Brundage, Peter Esselink and Andrew H. Baldwin conceived and designed the experiments, performed the experiments, analyzed the data, contributed reagents/materials/analysis tools, wrote the paper, prepared figures and/or tables, reviewed drafts of the paper.

Keryn B. Gedan analyzed the data, wrote the paper, reviewed drafts of the paper.

The following information was supplied relating to ethical approvals (i.e., approving body and any reference numbers):

USDA IACUC approval number: 103453.

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
