# Peer review of "Livestock as a potential biological control agent for an invasive wetland plant"

_PeerJ, doi:10.7717/peerj.567_

## Round 0.1 · original submission · Minor Revisions

This is a sound piece of work that hopefully will lead to practical applications (and incidentally replications in different systems).

Reviewer 1 ·

Basic reporting

I suggest that the authors consider integrating three points from the ecological literature in order to fully fit their work into the broader field of knowledge.

First, many experiments have examined the ability of herbivores to suppress or control invasive autotrophs. These have been formally synthesized by Levine et al. (2004) and Parker et al. (2006). Given the content of the current submission (i.e., large herbivore controlling an invasive autotroph), there seems to be a link to this part of the literature that should be made in the Introduction.

Second, and not mutually exclusive of the previous point, there have been studies on the degree to which native herbivores prefer exotic or native autotrophs (Mark Hay recent paper). This would also invoke ideas put forward by Cox and Lima (2006) TREE. In the current submission, it is interesting that, within N. America, an invasive autotroph is susceptible to an invasive herbivore. This suggests some relevance of historical exposure between herbivore and autotroph (both originating from Europe).

Finally, the Introduction indicates that most of the studies on large herbivores and invasive autotrophs have occurred in terrestrial grasslands; much of which show a weak effect. This is an important point, and the Introduction should better motivate why a different outcome should be expected in wetlands.

Experimental design

I find no fault with the experimental design and analysis.

A few questions:

(1) Why were 2 goats chosen as the herbivore density within the enclsoures?
(2) Did the duration of goat grazing intervals match practices of rotation grazing?
(3) Which part of the Phragmites was offer to the herbivores in the no-choice feeding assay? Does it matter if base, center, or top portions of plant are offered?
(4) In the grazed portion of the experimental enclosure, the percent cover of vegetation in the plot was ~50%. How does this compare to ungrazed, non-Phragmites marshes??

Validity of the findings

The findings are valid. But the Discussion (6.5 pages) seems relatively long compared to the rest of the paper (14 pages). In addition, much of the Discussion focuses on recommendations on how to implement (or not) livestock as a control mechanism for invasive autotrophs. But the current study did not test these recommendations. In addition, much of this is already touched on in Table 1.

I also recommend that the authors touch on the issue of ecological stoichiometry. Perhaps the seasonal decline in nutritional quality of Phragmites would enhance rather than diminish the consumption by goats so that the goats could meet their metabolic demand (C:N and C:P).

In figure 3A, it is unclear what the gray and black bars represent.

Additional comments

In general, I found this to be a very interesting and well-written manuscript. I look forward to seeing it in publication.

·

Basic reporting

This is an important paper and very relevant to decision makers concerned with conservation issues. It offers a real solution to controlling phragmites populations and also points out the caveats on how not to do it.

Experimental design

No comments

Validity of the findings

No comments

Additional comments

Win win solutions of economic gains in controlling invasives as proposed in this paper are often criticized with the argument that at some point the invasives are going to be needed to maintain the economic model based upon it. I don't think it is the case here as there are always other invasives for the animals to graze upon. Making this point though might ease some of the worries that people might have about this article. Choose as you please.

·

Basic reporting

I find this article to be very well written and have few comments. It is succinct, well-referenced, and clear. I like the comparative approach taken with Europe and find it encouraging that articles are being written that focus on management of invasives,such as Phragmites australis, rather than eradication.

Experimental design

The methods and experimental design are explained well and the data are presented clearly.

Validity of the findings

The use of livestock for controlling P. australis is clearly not a viable option everywhere, but the manuscript does a good job of presenting a clear benefit to livestock use while putting it in the context of where it can be an effective management tool.

Additional comments

In the legend for Fig 3B, line 536 refers to “same data as Fig. 4”. However, Figure 4 is photos – I’m not sure how to interpret this

The 2 photos in Fig. 4 look quite different from each other. They could use some explanation in the legend.

Reviewer 4 ·

Basic reporting

Line 124 – Rephrase “alternative plant prey options” – “alternative grazing options”
Line 126 – What is meant by a “pressed” experiment?
Line 133 - Goats should be identified by their scientific name or perhaps introduced in the methods as “domestic goats”
Lines 142, 163, and 164 - Spell out the numbers 1-9 unless used with units.
Line 153 – Missing a comma after stem density
Lines 185-186 – This statement could be clarified. What supporting data are you referring to? What other ways could native species have arrived at the site other than the two mentioned?
Line 266 – apostrophe is not needed in “Phragmites’”
Line 268-270 Sentence is unclear in context. Is the question which is more effective - springtime grazing or early summer grazing? If so, remove the word “even”.
Line 273 Change “those using invasive species control techniques” to “those using other invasive species control techniques”
Line 297 – I think this should be “in the Baltics”
Lines 330-332. The first sentence of the very last paragraph seems out of place as it relates mostly to the third-to-last paragraph. Consider revising this final paragraph.
Line 335 – Consider changing “from eradication to management” to “from eradication to mitigation”
Figure 4 includes two figures, A and B, but these are not referenced separately or given separate captions.

Experimental design

Lines 138-139. Is there a rationale for the design that goats are left in until Phragmites was within at least one of the four enclosures (vs. a set period of time)? Perhaps cite reference if this has been used before for grazing studies, or explain rationale for the design.

Figures 1 and 2. Grazed replicates had more desirable baseline measurements (e.g., Phragmites height and stem density is lower; species diversity and richness is higher), although the differences were not significant. However, it is worth clarifying in the text whether the plots were randomly allocated to the two study conditions.

Figures 1 and 2. The third grazing arrow aligns with measurements taken for % cover, species richness, and diversity. Please clarify in the figure whether the measurements were taken immediately before or after the grazing event for that timepoint.

Validity of the findings

Small # of replicates should be addressed as a limitation.

Additional comments

Were soil properties considered? Adding livestock to graze would affect nutrient cycling. May be beyond the scope of this paper but could be addressed in discussion.

Very interesting and well-written overall and I look forward to seeing more work in this area.

Reviewer 5 ·

Basic reporting

No Comments

Experimental design

The experimental design was appropriate, but there were only four replicates for each treatment. It is understood that this was a large-scale field experiment, so many replicates are not possible. Test statistics were not given, so I was unable to determine what the treatment effect size was, which would help to better determine if sample size was appropriate.

The design provided limited information about the potential of goat grazing to control Phragmites on a meaningful time scale. Does Phragmites density (cover) influence feeding preference of goats (alluded to but not directly tested)? What will happen after goats are completely removed?

Validity of the findings

Test statistics (F, MS, p, etc.) are essential if other studies are to be compared to this study.

A more realistic statement or discussion of the viability of this control approach is needed. I don't see how this type of grazing is a viable option given it is only a short term solution and Phragmites would either rebound or recolonize quickly after livestock removal, so benefits would only be short term. Can a native community, which serves some function, be maintained or will it be in some degraded state that is poor wildlife habitat? Most likely, the system will return to Phragmites dominance if restoration with competitive native species does not occur.

Additional comments

Introduction
Another commonly speculated reason for Phragmites decline in Europe is eutrophication (research by Armstrong and Armstrong).

Results
ANOVA statistics would be helpful.

Discussion
It appears from your results that Phragmites is high in protein, but has poor digestibility. Is this the case?

Line 270: remove underscore
Line 307: or not change at all.
Lines 309-311: How can long term livestock grazing be maintained without damage to native plants that are preferred or more palatable?
Line 324: data 'are' (plural)
Line 327: reducing instead of preventing?
Lines 328-329: Or where reduced P. australis biomass and some native cover is an acceptable goal.

---

## Round 0.2 · accepted · Accept

The changes made in this revised manuscript answer the comments made by the referees.

Note two minor issues: 1) please check carefully the reference list (I noted for example that the Parker et al. 2006 citation was not in the reference list, I guess it is Parker, J. D., D. E. Burkepile, and M. E. Hay. 2006. Opposing effects of native and exotic herbivores on plant invasions. Science 311:1459-1461?), 2) on page 10, when referring to fractional degrees of freedom, it is df, not ddf.